# Aquatic Bacteria *Rheinheimera tangshanensis* New Ability for Mercury Pollution Removal

**DOI:** 10.3390/ijms24055009

**Published:** 2023-03-05

**Authors:** Mengmeng Zhao, Gege Zheng, Xiuyun Kang, Xiaoyan Zhang, Junming Guo, Shaomei Wang, Yiping Chen, Lingui Xue

**Affiliations:** 1School of Biological and Pharmaceutical Engineering, Lanzhou Jiaotong University, No. 88, Anning West Road, Anning District, Lanzhou 730070, China; 2Northwest Institute of Eco-Environment and Resources, CAS, Donggang West Rd. 320, Lanzhou 730000, China; 3State Key Lab of Loess Science (SKLLQG), Institute of Earth Environment, Chinese Academy of Sciences, Xi’an 710061, China

**Keywords:** Hg-tolerant bacteria, extracellular polymeric substances, mer operon, dead bacterial biomass, environmental pollution remediation

## Abstract

To explore the strong tolerance of bacteria to Hg pollution, aquatic *Rheinheimera tangshanensis* (RTS-4) was separated from industrial sewage, with a maximum Hg(II) tolerant concentration of 120 mg/L and a maximum Hg(II) removal rate of 86.72 ± 2.11%, in 48 h under optimum culture conditions. The Hg(II) bioremediation mechanisms of RTS-4 bacteria are as follows: (1) the reduction of Hg(II) through Hg reductase encoded by the mer operon; (2) the adsorption of Hg(II) through the production of extracellular polymeric substances (EPSs); and (3) the adsorption of Hg(II) using dead bacterial biomass (DBB). At low concentrations [Hg(II) ≤ 10 mg/L], RTS-4 bacteria employed Hg(II) reduction and DBB adsorption to remove Hg(II), and the removal percentages were 54.57 ± 0.36% and 45.43 ± 0.19% of the total removal efficiency, respectively. At moderate concentrations [10 mg/L < Hg(II) ≤ 50 mg/L], all three mechanisms listed above coexisted, with the percentages being 0.26 ± 0.01%, 81.70 ± 2.31%, and 18.04 ± 0.62% of the total removal rate, respectively. At high concentrations [Hg(II) > 50 mg/L], the bacteria primary employed EPS and DBB adsorption to remove Hg(II), where the percentages were 19.09 ± 0.04% and 80.91 ± 2.41% of the total removal rate, respectively. When all three mechanisms coexisted, the reduction of Hg(II) occurred within 8 h, the adsorption of Hg(II) by EPSs and DBB occurred within 8–20 h and after 20 h, respectively. This study provides an efficient and unused bacterium for the biological treatment of Hg pollution.

## 1. Introduction

Mercury (Hg) is a common, naturally occurring, toxic heavy metal. It is easy to volatilize, spread, and stay in the atmosphere in the form of a gas, and redeposit in water or soil in the form of rain or dry gas. In this way, Hg already widely exists in the global environment and threatens the global biosphere [1]. Hg is extremely harmful to the environment because of its high toxicity, strong accumulation, and biological amplification. Various industrial wastewater discharges always contain Hg(II), such as from batteries, mining and smelting operations, industrial production and use, metallurgy, and electronics [2]. Hg(II) could accumulate in the kidney and cause acute renal failure, exposure to Hg(II) has become the main reason for autoimmune diseases and antinuclear antibodies. Hg(II) is also the pathogen of Alzheimer’s disease and Parkinson’s disease. Therefore, Hg(II) has a strong toxic effect on the central nervous system and digestive system [3]. Establishing efficient and green mercury remediation strategies has become an important issue in the field of environmental governance [4].

Many methods have been developed to remediate the pollution caused by Hg, such as the transference of Hg pollutants to remote areas, and removal through solidification and stabilization, cations, precipitation adsorption, and membrane filtration [5,6]. However, these approaches are usually costly, inefficient, and can easily produce by-products that cause secondary environmental pollution, unable to achieve the purpose of in situ remediation [7,8]. Microbial remediation is a safer, economical, and more effective alternative to traditional methods [9]. Sulfate-reducing bacteria were found to be very important in controlling both Hg methylation and MeHg degradation [10]. In 2022, it was found that the extracellular polymer of *Bacillus* could adsorb Hg, and its adsorption amount was 123.40 mg/g [11]; a strain of *Acinetobacter indicus* yy-1 was found to have a removal effect on both Cr and Hg [12]; *Pseudomonas* shows resistance to Hg and is able to transform Hg(II) to Hg(0), and shows great potential for the remediation of heavy metal-contaminated soil [13,14]. Many studies have showed that indigenous microorganisms found in the environment have a positive effect on the removal of Hg pollution, through reduction or adsorption [15,16,17], and current research mainly focuses on the isolation of microorganisms from soils. However, water is the main natural environment that accepts, degrades, and transforms various forms of Hg [5,18], so a water environment should also be a key focus for further research [19].

To explore the strong tolerance of bacteria for the bioremediation of Hg sewage, this study selected a bacterial strain of *Rheinheimera tangshanensis* (RTS-4), which has never demonstrated the ability to remediate Hg pollution from industrial wastewater, to explore its ability to tolerate and remove Hg(II). Then, the maximum Hg(II) tolerance concentration of the bacteria in solid and liquid media, optimal growth conditions, and Hg(II) removal efficiency were determined, and the mechanisms of Hg(II) removal by the RTS-4 bacteria were explored. Finally, the growth experiment of the indicator organism *Chlorella vulgaris*, verified the Hg(II) removal effect of the RTS-4 bacteria. The study found that RTS-4 bacteria can tolerate up to 120 and 60 mg/L of Hg(II), in solid and liquid media, respectively, and the removal rate of Hg can reach 86.72 ± 1.38%, at 48 h. RTS-4 bacteria also have the following capabilities: Hg(II) can be reduced by the mer operon during the 0–8 h period of bacterial growth; can be adsorbed by extracellular polymeric substances (EPSs) during the 8–20 h period, and can be adsorbed by dead bacterial biomass (DBB) after 20 h, until the treatment process is completed. This study provides valuable strain resources for the microbial remediation of Hg pollution, and suggests that the bacterium has broad application prospects in Hg pollution remediation.

## 2. Results and Discussion

### 2.1. Bacteria Identification and Hg Removal Ability

Ten bacterial strains with Hg-tolerant ability were isolated from a solid medium containing 120 mg/L Hg(II). One bacterial strain had 100% similarity to the *Rheinheimera tangshanensis* strain JA3-B52 (Figure 1A), and was identified as *Rheinheimera tangshanensis* (NCBI login number MT683260). This bacterial strain was termed RTS-4. Reports showed that bacteria in the genus of *Rheinoniae* generally play a role in plant growth and development, as rhizosphere microorganisms or endophytes [20,21]. The pectin removal property of *Rheinheimera tangshanensis* has substantial scope for its exploitation in wastewater treatment, and biopulping applications in the paper industry [22]. They can also produce DNA enzymes to mitigate the potential pleiotropic effects of new crop protection technologies, such as RNA interference [23]. However, few studies have demonstrated their strong Hg tolerance and conversion capacity, and it is rare for them to be used in the treatment of heavy metal wastewater.

The colony of RTS-4 was round and yellowish, with a neat edge and smooth surface, and the diameter of the colony was about 2–3 mm (Appendix A). Microscopic examination showed that it was comprised of rod-shaped Gram-negative bacteria (Figure 1B and Appendix A). The optimal growth conditions of RTS-4 were 30 °C, pH 7, 150 rpm, and 5% inoculation amount (Appendix A). Although these results are consistent with previous reports [17,24,25,26], our studies also found that the RTS-4 bacteria had good temperature and pH adaptability. They can grow well in the temperature range of 20–30 °C (Appendix A), exhibit high activity under neutral and acidic conditions (pH 6–7), and can grow normally under alkaline conditions (pH 8–9), which lays the foundation for the bacteria to demonstrate its bioremediation ability under various environmental conditions (Appendix A). Its maximum Hg tolerant concentrations in solid and liquid medium were 120 mg/L and 60 mg/L, respectively. Under optimal growth conditions, the Hg(II) removal rates of RTS-4 after 24, 48, and 60 h were 55.80 ± 1.30%, 86.72 ± 1.38%, and 86.98 ± 2.01%, respectively (Figure 1C). After 24 h, the Hg(II) removal rates of *Enterobacter helveticus* and *Brevundimonas* HgP1 were 28.80% and 63.60%, respectively, at a 5.5 mg/L initial Hg(II) concentration [27,28]. The Hg(II) removal rates of FG11 75B (*Enterococcus durans*), FG11 85F (*Enterococcus durans*), and AG0352A (*Enterococcus faecium*) were about 70% at a 5 mg/L initial Hg(II) concentration, after 48 h [29]. Compared with previous research, the RTS-4 bacteria still had good Hg(II) removal ability at higher concentrations of Hg(II) (10 mg/L), indicating its great potential for sewage treatment.

### 2.2. Hg Reduction Mediated by Mer Operon

After RTS-4 was activated by 10 mg/L Hg(II), the *mer*T gene showed expression after 2 h, reaching a maximum expression quantity after 6 h, and then downregulating from 8–24 h with negligible change (Figure 2B). After 2 h, the expression of *mer*T was about 428, 62, and 26 times higher than that of *mer*R, *mer*C, and *mer*A, respectively. After 6 h, it was about 17, 6, and 6 times higher than that of *mer*R, *mer*C, and *mer*A, respectively. The expression of *mer*R, *mer*C, and *mer*A genes showed similar trends, being significantly upregulated at 2 h, reaching a maximum expression after 6 h (Figure 2A,C,D), and showing downregulation after 8 h.

The mer operon mainly includes *mer*R, *mer*T, *mer*C, and *mer*A genes [15,29,30]. *mer*R is the promoter that initiates the expression of the entire operon. The *mer*T and *mer*C genes help Hg(II) transport from the outer membrane to the inner membrane and then deliver it to the Hg reductase binding site, which is encoded by *mer*A. Finally, Hg(II) is reduced to Hg(0) by Hg reductase [15,31].

In the genome of RTS-4, the *mer*R gene was upregulated at 2 h, but the expression level was downregulated due to the feedback inhibition of its own expression products after 8 h [15]. The *mer*T and *mer*C genes were significantly upregulated at 2 h, but became downregulated after 8 h, indicating that the bacteria began to regulate the transport of Hg(II) from outside to inside the membrane at 2 h but this ended after 8 h. The *mer*A gene changed from being significantly upregulated at 2 h, to downregulated after 8 h, indicating that Hg reductase began to synthesize and reduce Hg(II) to Hg(0) between 2 and 8 h, but its function ended after 8 h (the evolutionary trees of the four genes are shown in Appendix A). From the removal rate of Hg(II) by RTS-4 bacteria between 0 and 12 h, the reduction rate of Hg(II) after 2 h in the control group (without RTS-4 bacteria and adding 10 mg/L Hg(II)) and the experimental group (with RTS-4 bacteria and 10 mg/L Hg(II)) were 0.56 ± 0.01% and 0.54 ± 0.01%, respectively, indicating that the reduction of Hg(II) in the experimental group was caused by the volatilization of Hg; between 2 h and 8 h, compared with the control group, the removal rate of Hg(II) increased to 23.41 ± 0.93%, indicating that the bacteria had begun to remove Hg(II); this period is also when the mer operon plays its role. After 8 h, the bacteria still maintained a high Hg(II) removal rate from the adsorption of extracellular polymeric substances (EPSs) produced by the bacteria. Therefore, the removal rate of Hg(II) by the RTS-4 bacteria, at different times, can also confirm the role of the mer operon.

In the reported literature, the mer operon in *Staphylococcus* was significantly upregulated between 23 and 25 h. During this time, the expression of the *mer*T gene increased by about 9 times, compared to the control, and the *mer*R, *mer*C, and *mer*A genes could be activated by Hg(II) [15]. The mer operon in *Pseudomonas cremoricolorata* was significantly upregulated after 4 h, the expression of the *mer*T gene was much higher than that of the *mer*R, *mer*C, or *mer*A genes, and the Hg ion reduction process ended after 12 h [17]. The mer operon in *Pseudomonas caricapapayae* was expressed from 8 to 12 h, with the expression of the *mer*T and *mer*A genes being multiple times higher than that of the *mer*R and *mer*C genes [17]. Compared with the above bacteria, the expression times of the RTS-4 bacteria were early and short, showing that, as a newly discovered bacteria with Hg removal ability, the Hg removal mechanism of RTS-4 differed from that of the discovered bacterial strains. In addition, the expression level of the *mer*T gene was higher than that of the other genes, especially the *mer*C gene, which is also an encoded transporter. Therefore, the transporter encoded by *mer*T was mainly responsible for the transport of Hg(II) in RTS-4, similar to the findings of other reported results [17]. Comparing the action time of the mer operon with the point when RTS-4 reached the highest Hg removal rate, we believed that RTS-4 bacteria also contained other mechanisms for removing Hg pollution, to achieve the observed results.

### 2.3. Hg Adsorption Mediated by Extracellular Polymeric Substances (EPSs)

The content of the EPS polymer was 1025.46 ± 12.98 mg/L, indicating that RTS-4 could produce a large amount of the EPS polymer. Scanning electron microscopy (SEM) analysis showed that the cells were similar to the original bacterial strain in morphology (Appendix A), and did not produce mucus or filamentous substances after 10 mg/L Hg(II) treatment (Appendix A). However, when the concentration of Hg(II) was raised to 20 mg/L and 30 mg/L, the cell morphology remained intact but the number of deposits on the cell surface gradually increased (Appendix A). When the concentration of Hg(II) was raised to 40 mg/L, a large number of sediments appeared and some cells were sunken (Appendix A). We speculate that a large number of extracellular polymeric substances (EPS) began to be produced. At a Hg(II) concentration of 50 mg/L, the cell morphology was damaged, the surface was covered with mucus or filaments, and severe adhesion was observed. In addition to a large amount of EPSs on the surface of bacterial cells, dead bacterial biomass began to be produced (Appendix A). This indicated that the RTS-4 bacteria cells could grow very well in lower concentrations (10 mg/L) of Hg(II), gradually produced EPSs to adsorb Hg(II) at the concentrations of 20, 30, and 40 mg/L (medium concentrations), became damaged with the mass production of EPSs in high concentrations (50 mg/L) of Hg(II), and stopped growing and produced dead bacterial biomass in 60 mg/L Hg(II) (the maximum tolerance concentration of RTS-4 in liquid medium). Therefore, we believe that the concentrations of 10 and 50 mg/L are the thresholds for different mechanisms of bacterial Hg removal.

The adsorption rate of Hg(II) by EPSs reached 23.50 ± 0.98% after 120 h (Figure 3C). The adsorption rate of EPSs increased rapidly in the first 20 h and slowed afterwards (Figure 3C). Song et al., studied the effect of cyanobacteria extracellular polymer on Hg adsorption by goethite [32], while Dash and Das, isolated *Bacillus thuringiensis* and produced EPSs to adsorb Hg [29]. RTS-4 has a stronger ability to produce EPSs than the studied bacteria, under high concentrations of Hg(II) (50 mg/L) (Appendix A) [32,33,34].

After eliminating the influence of Hg(II) volatilization (Figure 3A), the adsorption rate of Hg(II) by EPSs between 0 and 20 h was found to account for 81.70 ± 1.33% of the total removal rate (Figure 3C). After 60 h, the adsorption rate gradually decreased (Figure 3B), indicating that the EPS removal of Hg(II) reached completion within 20 h.

Precipitation elemental analysis confirmed that there were five elements (carbon, nitrogen, oxygen, phosphorus, and Hg) on the surface of the cells (Figure 3E,F), and in the EPS solution (Figure 3G,H) containing 50 mg/L Hg(II). The proportions of C, N, O, P, and Hg were, respectively, 13.05 ± 0.84%, 15.49 ± 0.48%, 1.80 ± 0.07%, 6.80 ± 0.11%, and 49.13 ± 1.06% on the surface of the cells and 33.76 ± 0.98%, 14.19 ± 0.35%, 19.18 ± 0.12%, 8.38 ± 0.09%, and 24.10 ± 0.76% in the EPS solution. The SEM analysis, combined with the precipitation element analysis, showed that the EPSs produced by RTS-4 could embed Hg as spherical or amorphous sediments to achieve Hg removal. Francois et al. [33] isolated seven Hg-resistant strains, namely *Bacillus cereus*, *Lysinibacillus* sp., *Bacillus* sp., *Kocuria rosea*, *Microbacterium oxydans*, *Serratia marcescens*, and *Ochrobactrum* sp., that all produced EPSs. Inductively coupled plasma–optical emission spectroscopy (ICP–OES), and transmission electron microscopy (TEM) in conjunction with X-ray energy dispersive spectrometry, revealed that the bacteria incubated in the presence of HgCl_2_, sequestered Hg extracellularly, as spherical or amorphous deposits. However, no quantitative determination of the adsorption rate of EPSs for Hg(II) was found. The atomic percentage ratio of Hg in the precipitation formed by RTS-4 and EPSs was 2:1 (Figure 3F,H), indicating that, besides the EPSs reduction pathway, there is also a third Hg removal mechanism [34].

A series of changes in the functional groups of the EPSs before and after Hg(II) adsorption were characterized by FTIR, to further verify the adsorption characteristics of EPSs for Hg(II) (Figure 3D). The results indicated that Hg(II) adsorption by EPSs can be attributed to functional groups associated with proteins, lipids, and polysaccharides. Among them, the absorption peak at 3413 cm^−1^ shifted, indicating that the hydroxyl (–OH) group was involved in the adsorption of Hg(II) [35,36]. The C–H stretching vibration region shifted from 2938 cm^−1^ to 2936 cm^−1^, indicating that –CH_2_ was involved in the adsorption of Hg(II) [37]. The displacement from 1616 cm^−1^ to 1615 cm^−1^ indicated the stretching vibration of the double bond and the involvement of the C=O double bond, in Hg(II) adsorption [38]. The displacement at 1450 cm^−1^ signified the involvement of COO^−^ in Hg(II) adsorption [39], while the peak at 614 cm^−1^ was assigned to the symmetrical stretching vibration of the pyranose backbone [40]. The changes in the IR spectra observed in the 4000 and 500 cm^−1^ regions suggested changes in the –OH, –CH_2_, and –COO groups, which indicated that hydroxyl complexation and ion exchange occurred in EPSs during the adsorption of Hg(II).

### 2.4. Hg Adsorption Mediated by Dead Bacterial Biomass (DBB)

When an amount of 10, 20, 30, 40, 50, 60, 70, or 80 mg/L Hg(II) was added to the liquid medium, the adsorption rates of Hg(II) by DBB were respectively 41.80 ± 0.82%, 35.25 ± 1.24%, 23.17 ± 0.58%, 11.71 ± 0.26%, 12.98 ± 0.38%, 14.96 ± 0.51%, 14.59 ± 0.41%, and 13.17 ± 0.88% (Figure 4B). The correlation between the adsorption rate of Hg(II) by DBB and the concentration of Hg(II) in the solution, was determined by comparing the Hg(II) removal rates of the blank control and RTS-4 bacterial cells (Figure 4A) at different concentrations. The adsorption rate of DBB increased and then decreased (Figure 4B) with increasing Hg(II) concentration. At lower concentrations of Hg(II) (≤10 mg/L), RTS-4 bacteria rarely produce EPSs, and only reduce Hg(II) through mer operon and adsorption Hg(II) by DBB, which led to a higher adsorption rate of DBB [5]. When the concentration of Hg(II) increased to 40 mg/L, the EPS complex was produced in large quantities, and its adsorption was dominant, causing the adsorption rate of DBB to decrease.

Through calculation, the contribution rates of Hg(II) by DBB adsorption to the total treatment process were found to be 45.43 ± 1.02%, 38.47 ± 0.87%, 27.46 ± 0.74%, 17.25 ± 0.63%, 24.67 ± 0.82%, 27.63 ± 1.15%, 48.26 ± 1.31%, and 53.80 ± 1.47%, at the respective concentrations of Hg(II) (Figure 4C). The rate tended to increase, decrease, and increase again (Figure 4C). This trend is closely related to the production of DBB at different Hg(II) concentrations, and the mechanism of Hg removal by RTS-4. RTS-4 removes Hg(II) through Hg reduction by the mer operon, and Hg adsorption by DBB in the presence of Hg(II) concentrations equal to or lower than 10 mg/L; therefore, Hg adsorption by DBB accounted for a large proportion of the Hg removal process. When the Hg(II) concentration was between 10 and 50 mg/L (this concentration did not reach the maximum tolerance limit of RTS-4 bacteria), bacteria rarely produced DBB at this stage, and the three processes of Hg removal (i.e., mer operon reduction, EPS adsorption, and DBB adsorption) coexisted. Thus, the proportion of Hg(II) adsorption by DBB in the whole Hg removal process was lower. When the concentration of Hg(II) was above 50 mg/L, RTS-4 bacteria died in large numbers, and the adsorption process of DBB was dominant. This caused the proportion of DBB-adsorbed Hg(II) in the overall Hg removal process, to again increase [33].

### 2.5. Adsorption Kinetics

Adsorption kinetics were mainly used to analyze the relationship between the adsorption rate and adsorption time. This is an important parameter used to study the adsorption performance. The quasi-second-order kinetic model is based on the assumption that the adsorption rate is controlled by the chemical adsorption process [41,42]. In this experiment, we use the quasi-second-order kinetic model to describe the relationship between the amount of Hg(II) absorbed by the bacteria and time. The results showed that, when the initial concentration of Hg(II) was 10 mg/L, the adsorption of Hg(II) by RTS-4 was divided into two stages (Figure 4D): 0–2 h was a fast adsorption stage, during which the adsorption amount of Hg(II) reached 99.30 ± 2.58% of the total adsorption amount; the adsorption capacity of RTS-4 toward Hg(II) was 48.88 ± 2.83 mg/g. According to the parameters shown in Figure 4D, the fitting effect of the quasi-second-order kinetic equation was better, as the R^2^ was higher. The equilibrium adsorption quantity, *q_e_*, obtained from the quasi-second-order kinetic model was close to the equilibrium adsorption quantity measured after the experiment was completed; the pseudo-second-order kinetic model could more accurately describe the adsorption process of Hg(II) by RTS-4 bacteria. According to the mechanism of the quasi-second-order kinetic equation, it is inferred that the adsorption of Hg(II) by RTS-4 bacteria was a combination of physical and chemical adsorption. The chemical adsorption is dominant [43], which might be the result of the interaction between the functional groups on the cell surface and heavy metal ions [44]. When the initial concentration of Hg(II) was 10 mg/L, Kumar et al., showed that *chlorella* rapidly adsorbed Hg in the first hour, and reached the equilibrium adsorption capacity of 75.41 mg/g after 1 h [45]. Patiño-Ruiz et al., modified sodium alginate (Mat) microspheres with thiourea and magnetite nanoparticles, to adsorb Hg(II) ions in aqueous solution [36]. In the first 20 min, the adsorption rate of Hg(II) increased rapidly, accounting for 98% of the total adsorption amount, and reached equilibrium after 100 min (equilibrium adsorption capacity, 2.60 mg/g). A strain of arsenic-resistant bacteria, *Yersinia* sp., was also tested. The adsorption of SOM-12D on arsenic reached equilibrium at 60 min, and the equilibrium adsorption amount was 36.28 mg/g [46]. When the initial concentration of Cr(VI) was 50 mg/L, the adsorption capacity of the spores of *Aspergillus niger*, pretreated by freeze-thawing, increased rapidly in the first 25 min and reached the equilibrium adsorption state after 100 min, with an equilibrium adsorption amount of 40.63 mg/g [41]. Compared with the reported equilibrium adsorption amount, the equilibrium adsorption amount in this experiment was lower, which may be due to the self-metabolism of the bacteria used for adsorption and the different adsorption sites, or different bacteria having different repair mechanisms for Hg(II).

### 2.6. Hg Pollution Repair Mechanisms by RTS-4

In summary, Hg(II) removal by RTS-4 bacteria occurs through three mechanisms: (1) the reduction of Hg(II) by mer operon; (2) the adsorption of Hg(II) by EPSs; and (3) the adsorption of Hg(II) by DBB. When the environment contained lower concentrations of Hg(II) (≤10 mg/L), the RTS-4 bacteria mainly used the mer operon to reduce Hg(II), accompanied by the adsorption process of DBB, with contribution rates of 54.57 ± 1.41 and 45.43 ± 1.41%, respectively (Figure 5A). For moderate Hg(II) concentrations (10 mg/L < Hg(II) ≤ 50 mg/L), the RTS-4 bacteria first used the mer operon to reduce Hg(II), then focused on the production of EPSs to adsorb Hg(II), and finally formed DBB to adsorb any remaining Hg(II); the contribution rates of the three processes were 0.26 ± 0.01, 81.7 ± 0.74, and 18.04 ± 0.73%, respectively (Figure 5B). When the concentration of Hg(II) was >50 mg/L, the bacteria removed Hg mainly by producing a small amount of EPSs and a large amount of DBB to adsorb Hg(II), with the contribution rates of the two processes being 19.09 ± 0.44% and 80.91 ± 0.64%, respectively (Figure 5C). In chronological order, the mer operon reduction process was dominant between 0 and 8 h, the EPS adsorption process was dominant from 8 to 20 h, and the DBB adsorption process was dominant after 20 h (Figure 5D).

### 2.7. Verification of Hg Removal Effect

Some scholars have used *Chlorella vulgaris* (CV) as an indicator organism, to verify Hg(II) toxicity. Wang et al., evaluated the toxicity of Hg, Cu, Zn, Pb, and Cd toward CV, and found that Hg is most toxic to CV [5]. Duan et al., used a chlorophyll fluorescence analysis technique to study the combined toxic effects of Hg(II) on CV and to analyze the stress effect of Hg(II), using the chlorophyll content of CV [47]. Thus, we used *Chlorella vulgaris* (CV) and its chlorophyll content as an indicator to verify the Hg removal effect by the RTS-4 bacterial strain. The chlorophyll contents were 4.56 ± 0.08, 0.78 ± 0.02, 0.65 ± 0.03, 4.45 ± 0.12, 0.75 ± 0.03, and 4.13 ± 0.14 mg/L in CK, C1, C2, CK′, C1′, and C2′, respectively (Figure 6A). The chlorophyll contents changed little in CK, CK’, and C2′. However, the culture mediums in C1, C1′, and C2 turned yellow and precipitated, indicating low concentrations of chlorophyll and CV cell death (Figure 6B). After culturing for 36 h, the remaining concentrations of Hg(II) in the C1, C1′, C2, and C2′ solutions were, respectively, 9.47 ± 0.24, 9.35 ± 0.21, 9.23 ± 0.28, and 3.99 ± 0.07 mg/L. The removal rates of Hg(II) were 5.30 ± 0.11% (C1), 6.50 ± 0.09% (C1′), 7.70 ± 0.15% (C2), and 60.10 ± 2.12% (C2′).

In the experimental group (C1), where CV was added immediately after Hg(II) was added, a large amount of CV died within 12 h, due to Hg(II) poisoning. The culture solution had turned yellow and precipitated, indicating that the chlorophyll content was significantly reduced. Hg(II) volatilized for 24 h followed by the addition of CV (C1′), led to CV that was also poisoned by Hg(II), and a large number of deaths occurred within 12 h. This indicated the volatilization of Hg(II) could not alleviate its damage to CV. In the C2 group, where Hg(II), the RTS-4 bacterial strain, and CV were added at the same time, the CV still died within 12 h, causing the culture solution to turn yellow and precipitate, and the chlorophyll content to decrease. As the bacterial strain was unable to deal with the Hg(II) in time, CV still died in large numbers. Adding Hg(II) and RTS-4 bacterial strains 24 h prior to the addition of CV to the solution (C2′), caused most of the Hg(II) to be reduced or adsorbed by the bacterial strains. This allowed for the growth of CV to not be significantly affected, creating a solution that was green and clear (Figure 6).

The higher chlorophyll content, lower residual Hg concentration, and higher Hg removal rate of the C2′ experimental group, fully demonstrated the superior removal effect of the RTS-4 bacterial strain on Hg(II).

### 2.8. Research Prospects

The composition of industrial wastewater is complex and may contain many kinds of heavy metals. In this experiment, we found the tolerance and adsorption capacity of RTS-4 bacteria toward other heavy metals, and will discuss this more deeply in a followup report. In the future, we believe that immobilization technology can be used to study the process and control conditions of the immobilized bacteria for Hg-containing wastewater, to address the difficult reuse and poor stability of microbial remediation technology, and further improve its remediation efficiency. This research aims to provide a high-efficiency, environmentally friendly, and reusable approach for the biological treatment of Hg-containing wastewater.

## 3. Methods and Materials

### 3.1. Sampling and Purification

Wastewater was collected from the industrial sewage outfall in Xigu District, Lanzhou City, Gansu Province, China [48]. The enrichment, culture, separation and purification of bacteria were all performed as the methods described in Zhao et al.; the obtained bacterial suspension was stored in a refrigerator at −80 °C, for subsequent experiments [17].

### 3.2. Identification

To identify Hg-tolerant bacteria, 16S rRNA gene amplification was used. Using the common primers 27F and 1492R for the bacterial V3–V4 region [49], the amplification conditions and results detected were all performed as the methods described in Zhao et al. [17]. The sequencing results were submitted to the NCBI (www.ncbi.nlm.nih.gov/blast (accessed on 7 July 2020)) gene database to obtain the strain accession number MT683260. After BLAST comparison, we inferred the evolutionary history using the neighbor-joining method, and conducted evolutionary analyses in MEGA5 [50].

### 3.3. Optimal Growth Conditions

The single-factor analysis method was used to analyze the biomass of the Hg-tolerant strains under different Hg(II) concentrations, pH, temperatures, rotation speeds, and inoculation amounts, to determine the optimal growth conditions of the bacteria. For the preliminary experiment, we selected Hg concentrations of 0, 30, 60, 90, and 120 mg/L; pH values of 6, 7, 8, and 9; temperatures of 20 °C, 25 °C, 30 °C, 37 °C, and 42 °C; rotation speeds of 0, 50, 100, 150, 180, and 200 r/min; and inoculation amounts of 5%, 10%, 15%, and 20%. First, 1 mL of bacterial liquid was inoculated into 50 mL of LB liquid medium containing different Hg(II) concentrations, and cultured at 30 °C and pH 8 [33]. This was repeated in triplicate for each group. Culture medium (3 mL) was collected at regular intervals, and the absorbance (OD_600_) was measured with an ultraviolet–visible light spectrophotometer (UV-2100, Shanghai Unico Instrument Co., Ltd., Shanghai, China) to draw a growth curve. When selecting the optimal culture pH, temperature, rotation speed, and inoculum amount, 10 mg/L Hg(II) was added to the medium to measure bacterial growth, because the concentration of Hg(II) in general industrial wastewater did not exceed 10 mg/L [51,52,53].

### 3.4. Removal Rate

One milliliter of logarithmic growth phase bacterial culture medium was added to 50 mL of LB liquid medium containing 10 mg/L Hg(II), and cultivated at 30 °C and 150 rpm, using the LB liquid medium containing 10 mg/L Hg(II), without bacteria, as a control. An atomic fluorescence Hg meter Mercur^®^ (AFS, Jena Analytical Instruments Co., Ltd., Jena, Germany) was used to detect the remaining concentration of Hg(II) after treatment at different times (24, 48, and 60 h), and the removal rates of Hg(II) were calculated by the following equation:(1)$$C=\frac{C_{0}-C_{1}}{C_{0}}\times 100\%$$
where *C* is the Hg(II) removal rate, *C*_0_ is the initial Hg(II) concentration in the samples, and *C*_1_ is the remaining Hg(II) concentration in the samples after bacterial treatment.

### 3.5. Growth Curve

Fresh bacterial liquid was inoculated into 50 mL of LB liquid medium, according to the optimal inoculation amount selected in the previous experiments, under the conditions of 0 and 10 mg/L Hg(II). The biomass (OD_600_) of the bacterial solution was measured initially, and sampling every 2 h until 12 h. After 12 h, the biomass (OD_600_) of the bacterial solution was measured every 12 h. The growth curves of the strains were plotted, and changes in the growth curves of the strains with and without Hg(II) were compared.

### 3.6. RT-qPCR

To activate the bacteria, 10 mg/L of Hg(II) was added to the LB liquid medium, and the liquid medium without Hg(II) was used as a control. To study the real-time changes in gene expression in the mer operon, according to the results of preliminary experiments, measurements taken at 2, 6, 8, 12, and 24 h were selected for the determination of gene expression. The RT-qPCR experiment detected four main genes in the mer operon, *mer*A, *mer*R, *mer*C, and *mer*T, and the whole process was conducted on ice. The specific genes and 16S rRNA housekeeping gene primers used were designed and synthesized by Lanzhou Ruizhen Biotechnology Co., Ltd., Lanzhou, China (the primers of the *mer*A, *mer*R, *mer*C, *mer*T genes, and the housekeeping gene, are provided in Appendix A). The 16S rRNA house-keeping genes were used as the internal reference gene, and the *Ct* value method was used to calculate the expression level of each gene:F = 2*^−^*^∆∆*Ct*^(2)
where ∆∆*Ct* = (the *Ct* value of the target gene in the experimental group − the *Ct* value of the internal reference gene in the experimental group) − (the *Ct* value of the target gene in the control group − the *Ct* value of the internal reference gene in the control group). Therefore, 2^−∆∆*Ct*^ represents the fold change in the target gene expression in the experimental group relative to the control group.

### 3.7. Extracellular Polymeric Substances (EPSs)

The bacterial strains were collected by centrifugation in a culture medium containing 50 mg/L Hg(II), and observed under a scanning electron microscope (TESCAN MIRA3, Brno, Czech Republic), at a magnification of 15.0 KX and a voltage of 8.0 kV.

The experimental process of obtaining EPSs was referred to Francois et al. [33], the EPSs produced by RTS-4 were extracted by high-speed centrifugation and were identified using a TOC instrument. The precipitation of EPSs and Hg(II) in the dialysis bag was vacuum freeze-dried; one portion was directly used for scanning electron microscopy, and the other was used for scanning energy dispersive spectroscopy (EDS) analysis.

### 3.8. Analysis by Fourier Transform Infrared (FTIR) Spectroscopy

The extracellular polymeric EPS purified sample was freeze-dried (FreeZone 6, Labconco Corporation, Kansas City, MO, USA), the dried sample was ground into a powder, and the slices were made via a KBr Fourier transform method and analyzed by infrared spectroscopy (Vertex 70, Bruker Spectroscopy, Ettlingen, Germany).

### 3.9. Dead Bacterial Biomass (DBB)

Preparation of dead bacteria was according to a previous study [54]. The dead bacterial cells were placed in media containing 10, 20, 40, 60, and 80 mg/L Hg(II), respectively, with the blank medium and medium with live bacteria used as controls. The contribution rate at different concentrations of dead bacteria, toward the adsorption of Hg during heavy metal removal, was analyzed. An AFS was used to detect the remaining concentration of Hg(II) in the culture medium after 0 h and 120 h treatment, and the Hg(II) adsorption rate was calculated using Equation (1).

### 3.10. Adsorption Kinetics Experiment

The suspension of RTS-4 bacteria from the logarithmic stage was cultured to 2 mL. After centrifugation, 0.1 g of bacteria was added to 50 mL of distilled water containing 10 mg/L Hg(II), and was oscillated and adsorbed at 30 °C in a constant-temperature shaking bed, at 150 rpm. The residual Hg(II) concentrations were determined by AFS at 0.5, 1, 2, 4, 6, 8, 12, and 24 h, and the adsorption kinetic curve was drawn.

The pseudo-second-order kinetic equation can be written as
(3)$$\frac{t}{q_{t}}=\frac{1}{K_{2}q_{e}^{2}}+\frac{t}{q_{e}}$$
where *q_t_* is the adsorption capacity at time *t* (mg/g), *q_e_* is the equilibrium adsorption capacity (mg/g), *t* is the adsorption time (h), and *K*_2_ is the pseudo-second-order adsorption rate constant.

### 3.11. Verification by Bio-Indicator

The growth status of *Chlorella vulgaris* (CV) was used to determine the Hg(II) reduction abilities of Hg-tolerant bacterial strains. CV was originally cultured in a BG11 medium [55]. The specific experimental procedures was referred to Zhao et al. [17].

### 3.12. Statistical Analysis

In this study, the SPSS 20.0 software (IBM SPSS Statistics for Windows, version 22.0, IBM Corp., Armonk, NY, USA) was used for the analysis of all data, the Origin 9.0 software (version 9.0, OriginLab, Northampton, MA, USA) was used for chart processing, and the phylogenetic tree was constructed using the MEGA5 bioinformatics software [50].

## 4. Conclusions

In this study, a bacterial strain of RTS-4, with strong tolerance to Hg(II), isolated from industrial wastewater, was used to analyze the Hg pollution repair ability and mechanism. It was found that the bacteria can reduce Hg(II) to Hg(0) by Hg ion reductase, to alleviate the toxicity of Hg(II), and adsorb Hg(II) by producing extracellular polymers and dead biomass to reduce the concentration of Hg(II) in the environment. When different concentrations of Hg(II) are found in the environment, these three mechanisms work alone or in synergy, to remediate Hg pollution in the environment. The effect of Hg(II) remediation by RTS-4 was verified through a *chlorella* growth experiment. Because this kind of microorganism (*Rheinheimera tangshanensis*) has high tolerance, and a strong biotransformation ability and adsorption ability towards Hg, and has never before been found to have the ability to repair Hg(II) pollution, this study broadens the source of bacterial strains for Hg pollution bioremediation, provides valuable bacterial strain resources, and introduces a theoretical basis for the application of this bacterial strain in Hg pollution treatment in water.

## Figures and Tables

**Figure 1 ijms-24-05009-f001:**
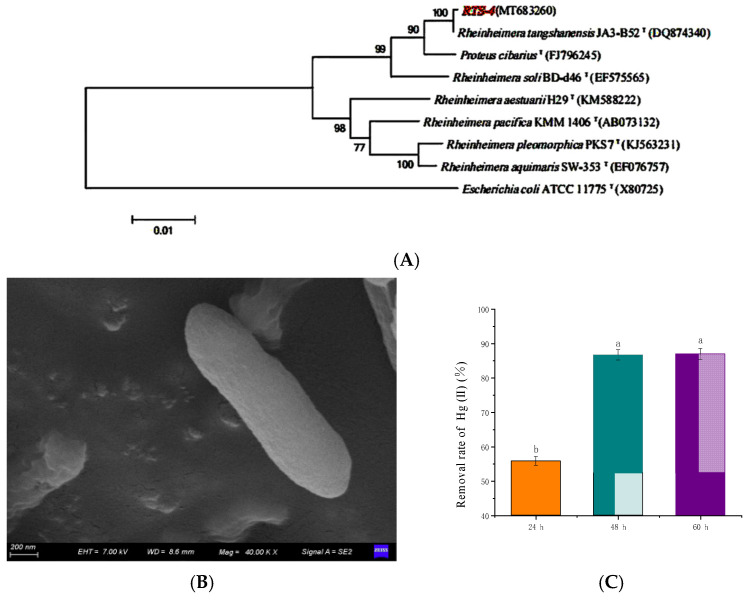
(**A**) Gene sequence phylogenetic tree of the RTS-4 bacteria. Taxonomic classifications were assessed by comparison with the NCBI nucleic acid database. Numbers denote closeness distances between adjacent species. The evolutionary history was inferred using the neighbor-joining method. Escherichia coli ATCC 1175T(X80725) was used as an outgroup. The bootstrap consensus tree inferred from 1000 replicates is taken to represent the evolutionary history of the taxa analyzed. Branches corresponding to partitions reproduced in less than 50% of bootstrap replicates are collapsed. The percentage of replicate trees in which the associated taxa clustered together in the bootstrap test (1000 replicates) are shown next to the branches. The tree is drawn to scale, with branch lengths in the same units as those of the evolutionary distances used to infer the phylogenetic tree. The evolutionary distances were computed using the maximum composite likelihood method, and are in the units of the number of base substitutions per site. The analysis involved 24 nucleotide sequences. Codon positions included were 1st, 2nd, 3rd, and noncoding. All positions containing gaps and missing data were eliminated. There were a total of 1170 positions in the final dataset. Evolutionary analyses were conducted in MEGA5. (**B**) Scanning electron microscopy (SEM) photograph of the RTS-4 bacteria (Voltage, 7.00 kV; magnification, 40.00 KX). (**C**) Hg(II) removal rate of the RTS-4 bacteria in liquid medium at different times. Bars (mean values ± SD) with different letters are significantly different from the three independent experiments at the 0.05 level, according to LSD multiple comparisons.

**Figure 2 ijms-24-05009-f002:**
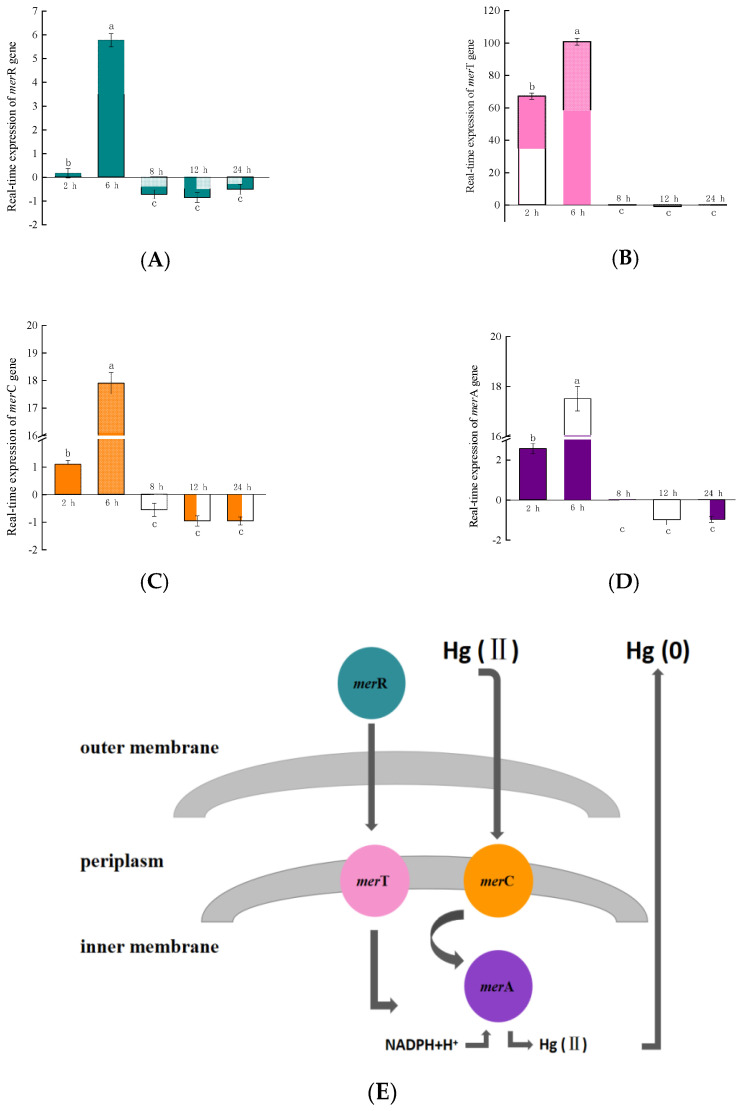
Real-time quantitative analysis of the principal genes, (**A**) *mer*R, (**B**) *mer*T, (**C**) *mer*C, and (**D**) *mer*A, in the mer operon of the RTS-4 bacterial strain at different times. (**E**) Hg(II) reduction mechanism in the RTS-4 bacterial strain. Bars (means values ± SD) with different letters are significantly different from the three independent experiments at the 0.05 level, according to LSD multiple comparisons.

**Figure 3 ijms-24-05009-f003:**
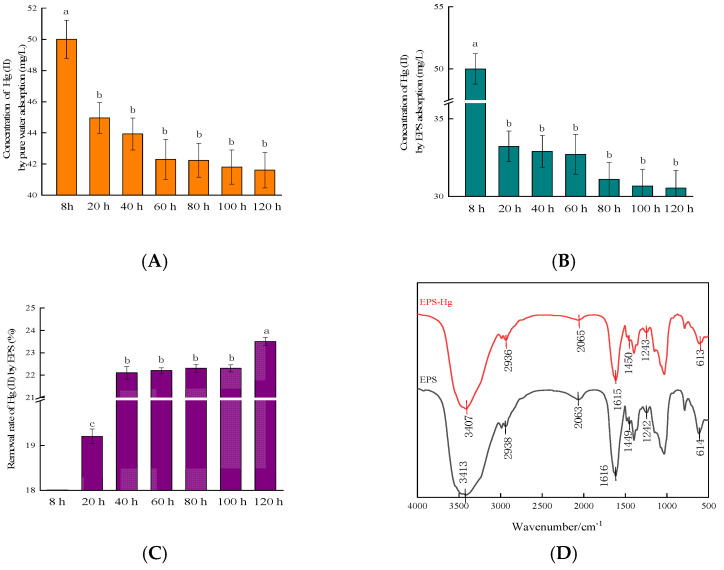
Analysis of Hg(II) adsorption by extracellular polymeric substances (EPSs) produced by the RTS-4 bacteria. (**A**) Residual Hg concentration in the solution after pure water adsorption in the dialysis bag at different times (control group); (**B**) residual Hg concentration in the solution after EPS adsorption in the dialysis bag at different times; (**C**) removal rate of Hg(II) by EPSs at different times; (**D**) infrared spectrometric analysis of extracellular polymeric substances (EPS) cultured in liquid medium with (red line) and without (black line) Hg(II) (the numbers indicate the peak positions of the functional groups); (**E**) scanning electron microscopy (SEM) photograph of the surface of the RTS-4 bacterial cells that were cultured in the liquid medium containing 50 mg/L Hg(II); (**F**) precipitation element analysis by energy dispersive spectroscopy (EDS) on the surface of the RTS-4 bacterial cells that were cultured in the liquid medium containing 50 mg/L Hg(II); and (**G**) SEM photograph of the surface of the EPSs that were produced by RTS-4 bacteria in the liquid medium containing 50 mg/L Hg(II). (**H**) Precipitation element analysis by EDS on the surface of the EPSs that were produced by the RTS-4 bacteria in the liquid medium containing 50 mg/L Hg(II). Bars (means values ± SD) with different letters are significantly different from the three independent experiments at the 0.05 level, according to LSD multiple comparisons.

**Figure 4 ijms-24-05009-f004:**
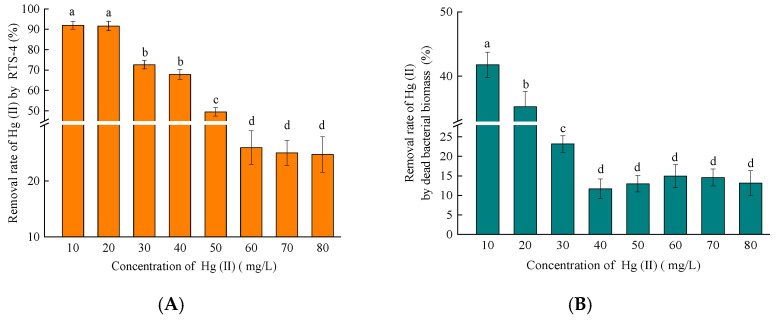
Analysis of Hg(II) adsorption by the dead bacterial biomass (DBB) produced by the RTS-4 bacteria under different concentrations of Hg(II). (**A**) Removal rate of Hg(II) by the RTS-4 bacterial strain under different concentrations of Hg(II) (control group); (**B**) removal rate of Hg(II) by the dead bacterial biomass produced by the RTS-4 bacterial strain under different concentrations of Hg(II); (**C**) dead bacterial biomass removal rate compared to the total removal rate by the RTS-4 bacterial strain; and (**D**) biosorption dynamic curves of RTS-4 bacteria. *q_t_* represents the amount of adsorption at t time. “Second kinetic” represents a second-order kinetic curve. The table in Figure D shows the second-order kinetic parameters of RTS-4. *q_e_* represents the adsorption capacity at t time (mg/g); and *K*_2_ represents the pseudo-second-order adsorption rate constant; R^2^ represents the goodness of fit. Bars (means values ± SD) with different letters are significantly different from the three independent experiments at the 0.05 level, according to LSD multiple comparisons.

**Figure 5 ijms-24-05009-f005:**
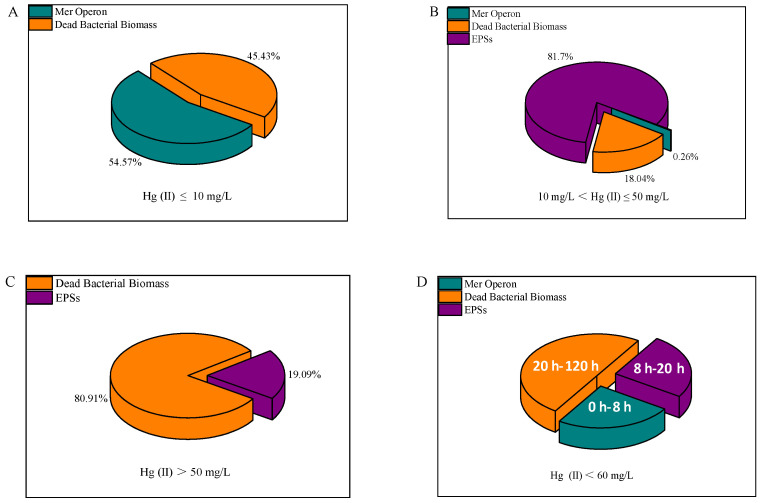
Overall removal mechanisms of Hg(II) by the RTS-4 bacteria, including the removal mechanisms when (**A**) Hg(II) ≤ 10 mg/L, (**B**) 10 mg/L ≤ Hg(II) ≤ 50 mg/L, and (**C**) Hg(II) > 50 mg/L. (**D**) Principal removal mechanism at different times. “Dead bacterial biomass” represents the adsorption of Hg (II) by dead bacterial biomass; “Mer operon” represents the reduction of Hg(II) by mer operon, and “EPSs” represents the adsorption of Hg(II) by producing extra-cellular polymeric substances (EPSs).

**Figure 6 ijms-24-05009-f006:**
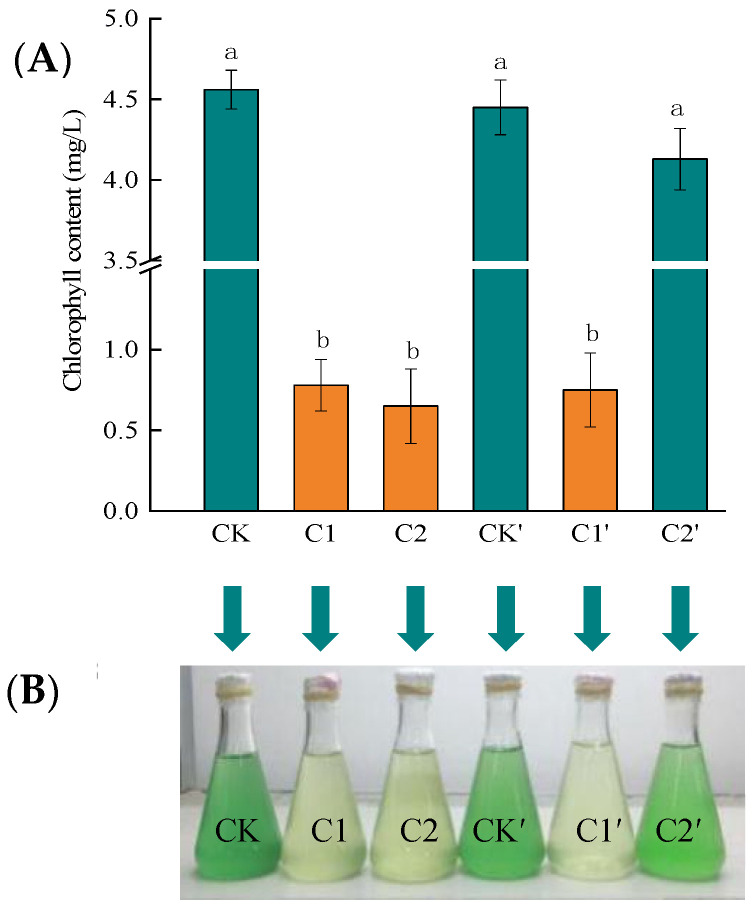
Treatment effects of the RTS-4 bacterial strain on Hg-containing sewage. (**A**) *Chlorella vulgaris* (CV) without any treatment (CK); addition of CV after 24 h (CK′); addition of CV treated with 10 mg/L Hg(II) (C1); addition of CV after 24 h treatment with 10 mg/L Hg(II) (C1′); addition of CV treated with 10 mg/L Hg(II) and RTS-4 Hg-tolerant bacteria (C2); and addition of CV after 24 h treatment with 10 mg/L Hg(II) and RTS-4 Hg-tolerant bacteria (C2′). The CV and Hg-tolerant bacteria added to each experimental group were 20% and 6% at concentrations of 3 × 10^7^ cells/mL and 10^9^ CFU/mL, respectively. (**B**) CV growth effects on Hg-containing sewage in different treatment groups. Bars (mean values ± SD) with different letters show significant differences at the 0.05 level, according to the test or multiple LSD comparisons from three independent experiments.

## Data Availability

Not applicable.

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
