# Peer review of "Aquatic Bacteria Rheinheimera tangshanensis New Ability for Mercury Pollution Removal"

_ijms, 2023, doi:10.3390/ijms24055009_

Round 1

Reviewer 1 Report

Recommended for the publication

Reviewer 2 Report

Dear authors,

Congratulations on your work and on your decision to publish the results you achieved. Before publishing, I have some recommendations for you:

1. Could you please elaborate on the abstract's last sentence and emphasize the advantages? "This study provides an efficient and unused bacterium for the biological treatment of mercury pollution." 

2. Introduction lines 40 - 42, need to be revised, please find a synonymous to "to repair" it is not the meaning you meant.

3. Lines 63-67 need to give more insight data and results.

"Based on the described mechanism for the removal of Hg(II) pollution in water by RTS-4, this study provides valuable strain resources for the microbial remediation of mercury pollution and suggests that the bacterium has broad application prospects in mercury pollution remediation." - could you detail and reason your conclusions?

4. The Introduction section need to be more elaborated and consider more scientific papers published on the same topic in the last 3 years, no older than. Pay attention there are already papers published for 2023 and you could cite them as well.

5. The next section should be methodology/materials and methods used to develop your experiments/study, then you need to present the experimental data, and afterward you can present the results, discussion, and conclusions. Section 2 in your case needs to be renamed, restructured, rewritten, and reorganized, you do not present herein the results that you developed within your experiments, it is more of literature research, isn't so?

6. References are not according to the instructions, please pay attention to this issue as well.

7. I would have expected to see the adsorption kinetic curves. Could you complete with? 

8. Conclusions section need also to be reconsidered and give more insights into the technical results that you achieved through your experiments and what are the contributions of your findings to the scientific, academic, or practical studies.

Good luck with your work!

Reviewer 3 Report

General comments

The paper by Zhao et al. represents a good, comprehensive, interesting, and valuable study of the mercury pollution repair ability and mechanism of a bacterial strain with strong tolerance to Hg(II) isolated from industry wastewater. 

It is clearly written, well referenced and the results are well elaborated and documented.

All aspects are discussed in detail. This study is important for the ongoing research on bioremediation treatment process through mercury-resistant bacteria. 

I have only few suggestions for revising the manuscript.

Specific comments

Page 1, line 2 and lines 12-13. Please italicize Rheinheimera tangshanensis.

Page 2, lines 33-34. I suggest rephrasing the sentence “It can exist for long periods in the environment and migrate widely around the world through water and air” explaining that mercury can be emitted in the environment from natural and anthropogenic sources. Differently to other metals, it evaporates easily and exists in the atmosphere almost in the gaseous phase. Once emitted, it can travel long distances in the atmosphere where it can remain for more than a year before it is eventually deposited back to water and soil surfaces in rainfall or in dry gaseous form. In this way, mercury has become widespread throughout the worldwide environment and threatens the biosphere globally.

Moreover, I suggest mentioning how mercury pollution affect human health.

Page 2, line 48-49. I suggest specifying in which year “Pseudomonas mutants were found to be able to re- mediate mercury-contaminated soil”, in order to conform to the other citations.

Page 3, line 126. Please write respectively at the end of the sentence “After 6 h, it was 125 about 17, 6, and 6 times higher than that of merR, merC, and merA.”

Page 5, lines 179-180. I suggest moving the sentence “The EPSs produced by RTS-4 were extracted by high-speed centrifugation and were identified using a TOC instrument.” in the Methods and materials section.

Page 8, line 279. I suggest to replace “of lower concentrations of Hg(II) (≤10 mg/L)” with “of Hg(II) concentrations equal or lower than 10 mg/L”.

Page 9, line 321. “chlorella” is not italicized.

Page 13, line 496. There is a typo: “heavy metal” at the beginning of the line.

Figure 3. The figure 3D is not mentioned in the text.

Finally, I suggest standardizing the way of indicating mercury throughout the text. Sometimes I found "mercury", other times “Hg”.

Round 2

Reviewer 2 Report

Dear authors,

Congratulations for your work and thank you for considering all my recommendations.

There is still one more that you need to do - to change the title and replace "repair" with "removal".

My suggestion is: Aquatic bacteria Rheinheimera tangshanensis new ability for mercury pollution removal

Good luck!
